# Sorting Out the Risks and Benefits of the #797 Recommended Intrapartum Vancomycin Dosing Approach

**DOI:** 10.3390/antibiotics12010032

**Published:** 2022-12-25

**Authors:** Andras Farkas, Arsheena Yassin

**Affiliations:** 1Optimum Dosing Strategies, Bloomingdale, NJ 07403, USA; 2Department of Pharmacy, Saint Clare’s Health, Denville, NJ 07834, USA; 3Department of Pharmacy, Robert Wood Johnson University Hospital, New Brunswick, NJ 08901, USA

**Keywords:** intrapartum, pharmacokinetics, vancomycin

## Abstract

ACOG Committee Opinion #797 proposed intrapartum vancomycin dosing guidelines in the absence of thorough evaluation of its risk versus benefit profile on the maternal and neonatal systems. The previously published serum and cord-blood concentration–time data of vancomycin given to mothers in the intrapartum period was analyzed in this work with a two-compartment pharmacokinetic (PK) model. Monte Carlo simulation was used to establish exposure for the studied population for doses of 1000 mg to 2000 mg every 8 h for gestational ages (GA) of 33 to 40 weeks and for birth times up to 4-h intervals. Probabilities of target attainment (PTA) were calculated for efficacy and toxicity indices unique to the peripartum maternal and neonatal population. Neonatal evaluations indicate uniformly high PTAs for the evaluated dosing regimens when the efficacy target is considered. On the other hand, the PTAs for potentially nephrotoxic exposure is expected to reach undesirable levels when three or more doses were to be administered. The risk is profoundly high in GA below 36 weeks and birth times beyond 20 h after the initiation of intrapartum prophylaxis and with doses greater than 1250 mg. Maternal vancomycin exposures seem reasonable up to two intrapartum doses given at 8 h intervals when the dose is kept to 1250 mg or less. Most mothers (up to 83%) who receive three or more doses of the commonly administered regimens are subjected to nephrotoxic exposures. Thus, it appears that the current recommendations by #797 for dosing of vancomycin pose considerable risk to mother and newborn alike, especially in cases with lengthy duration of preterm labor. Capping of doses at 1250 mg may be considered to minimize the need for therapeutic drug monitoring (TDM) interventions. Alternatively, and irrespective of the baseline maternal renal function, TDM for all cases requiring more than two doses of 1500 mg or higher must be assured.

## 1. Introduction

According to the American College of Obstetricians and Gynecologists (ACOG), all pregnant women should undergo prenatal screening for Group B streptococcus (GBS) [1]. Although GBS is a physiologic component of the intestinal and vaginal microbiome, it may cause systemic maternal infections and is associated with stillbirth and preterm labor. GBS is also the leading cause of newborn infections. To prevent newborn diseases, intrapartum prophylaxis is recommended for GBS-positive women. Penicillin is the agent of choice, and ampicillin is an alternative. For women with allergy history suggesting a low risk of anaphylaxis to penicillin, the first-generation cephalosporins are indicated. Alternatively, for high risk of anaphylaxis and with history of severe reactions, intrapartum prophylaxis with clindamycin and vancomycin are suggested. In #797, ACOG published updates on intrapartum vancomycin dosing for prophylaxis, now recommending 20 mg/kg IV every 8 h with a maximum single dose of 2000 mg [1]. These new dosing guidelines are based mainly on data from two papers, where the success of achieving therapeutic vancomycin exposure was met when the measured maternal and cord blood concentrations fell anywhere within the range of 10 mg/L and 40 mg/L, irrespective of the total systemic exposure [2,3,4]. The aim of this study was to analyze the published data from Towers et al. (2017) in a contemporary pharmacometric context and to raise awareness about the risks associated with the proposed dosing approach. Further, we also aim to provide alternative dosing concepts that allow clinicians to provide safe and effective intrapartum prophylaxis practices.

## 2. Results

The demographic, clinical, and vancomycin dosing and laboratory characteristics of the 30 mother–newborn pairs who participated in the study have been published elsewhere. The population PK model parameter estimates for intrapartum vancomycin are provided in Table 1 [3]. The individual observed versus predicted concentrations showed good model fit (Figure 1). The analysis failed to select for maternal body weight as covariate to be included in the final model.

**Table 1 antibiotics-12-00032-t001:** Pharmacokinetic parameter estimates.

Parameter	Median (%CV)
V, maternal (L)	39.66 (49.9)
V, fetal (L)	2.07 (39.5)
CL, maternal (L/h)	4.78 (43.6)
K_mf_ (h^−1^)	0.51 (31.7)
K_fm_ (h^−1^)	0.36 (34.1)

For description of model parameters, see Figure 2.

**Figure 1 antibiotics-12-00032-f001:**
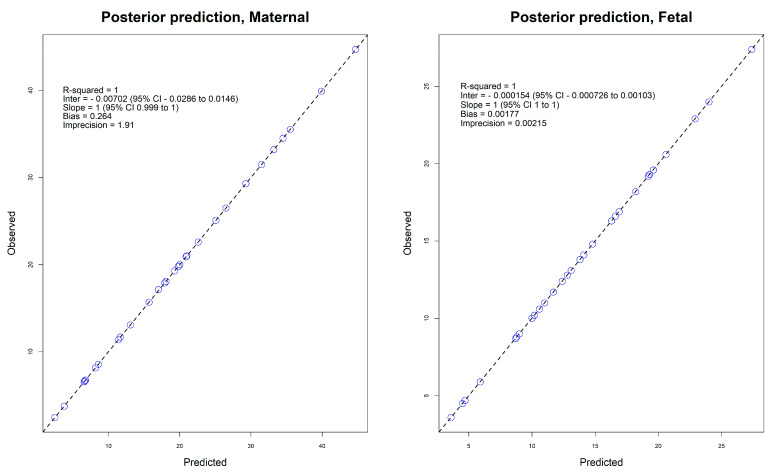
Plots of observed and predicted posterior individual serum concentrations.

**Figure 2 antibiotics-12-00032-f002:**
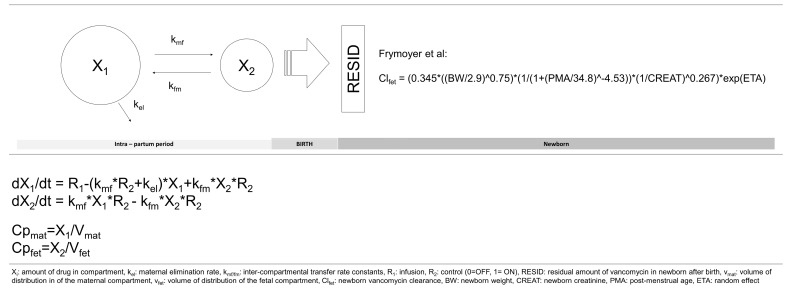
Intrapartum intravenous vancomycin model structure and equations [5].

PTA attainment analysis results for each regimen, GA, and birth time for the neonate and the mother are presented in Figure 3, Figure 4 and Figure 5. For the efficacy target, simulations for the neonates suggest greater than 90% PTA at the pre-specified target when the intrapartum prophylaxis is commenced at least 4 h before the time of birth, whereas when initiated within the first 4 h, the target attainment is evenly limited and shows no evidence of a distinctly superior dosing approach for the studied population and among the unique dosing regimens evaluated (Figure 3). When compared with the maximum recommended empiric neonatal vancomycin dose for the first day of life, standardized maternal doses of 1250 mg or less consistently show the magnitude of total drug exposure (3750 mg or less total daily dose) that minimizes the risk of neonatal overdosing when no more than three intrapartum doses are administered. As the maternal dose increases to 1500 mg on day 1, up to 23.1% of preterm (GA < 36 weeks) neonates are likely to accumulate the drug in excess of the 30 mg/kg/day dose (most at risk when GA < 34 weeks and time of birth 18 h or later). For the dose of 1750 mg, up to 28.3% of preterm (GA < 36 weeks) neonates are likely to accumulate the drug in excess of the 30 mg/kg/day dose (most at risk when GA < 35 weeks and time of birth > 18 h), while for the dose of 2000 mg, the value increases up to 34.0% (most at risk when GA < 35 weeks and time of birth > 12 h). If additional maternal doses are administered on day 2 in the preterm population, then the proportion of neonates at risk of overdose will continue to grow to between 10% and 20% (1000 mg) to beyond 30% (2000 mg), irrespective of the dosing regimen. For GA of 37 weeks or more (early and full-term), only doses of 1250 mg or less showed consistently low risk (1–10%) of overdose on day 2.

Daily AUC-exposure-driven probability of nephrotoxicity assessment indicates excess risk with the 2000 mg on day 1 when three doses were administered, regardless of the GA (Figure 4A) and at both target values evaluated. In case of a prolonged labor necessitating 4 or more intrapartum doses to be administered, the 1250 mg dose assures that only a small fraction of the neonatal population would be subjected to nephrotoxic exposures with risk greater than 10% realized after five prophylactic doses are given for cases where birth will occur 36 h post prophylaxis commencement and predominantly in the preterm population (Figure 4B). As the dose increases from 1500 to 2000 mg, then the proportion of early and full-term populations born 30 h or later with risk beyond the cutoff of 10% also begins to grow to a maximum value of 29.7% and 7.9% for the 1500 mg dose, 44.3% and 16.1% for the 1750 mg dose, and 58.2% and 29.9% for the 2000 mg dose for the targets of 600 and 800 mg·h/L, respectively (Figure 4C–F). The proportion of newborns expected to surpass the trough level of 15 mg/L prior to the fourth dose by more than 10% is foreseeable for all regimens with doses equal to and greater than 1250 mg.

For the toxicity index of the AUC > 600 mg·h/L, simulation results show early signs of excessive drug exposure in maternal serum for the population as a whole and all dosing schemes beyond 1250 mg with probabilities surpassing 27.6% for doses of 1500 mg or greater on day 1 and with the administration of three doses. On day 2, incremental increase in the chance of toxic maternal exposure is expected with subsequent doses given, showing the greatest risk at 83.9% with the current guideline recommended maximum dose of 2000 mg (Figure 5).

## 3. Discussion

A method unique to our analysis is the use of population PK, which enabled us to accurately describe the concentration–time profile together with the corresponding drug exposure that in turn allows for Monte Carlo simulation of competing dosing regimens and assessment of the PTA for efficacy and toxicity indices on both the maternal and newborn systems. Simulation results confirm robust target attainment for the #797 proposed dosing regimen at the efficacy index of the AUC/MIC ratio of 39.1 but considerably lower success rates when the toxicity targets of AUC > 600/800 mg·h/L, trough >15 mg/L, or the 30 mg/kg/day maximum neonatal treatment dose for the first day of life are considered and especially when used in mothers with prolonged preterm labor. Our analysis confirms that standardized intrapartum doses of 1250 mg given at 8 h intervals provide maximally safe and sufficiently effective exposure for both the maternal and neonatal population and that further increases in the dose to a maximum of 2000 mg result in no meaningful gain in efficacy target attainment, irrespective of the GA or the time of birth after the commencement of intrapartum prophylaxis, and would require close TDM.

Two main reasons for the need to administer such aggressive dosing regimens are cited by #797.

The focus of the first rationale is centered on the assumption of the lack of adequate neonatal vancomycin exposure provided by the “at that time” standard dose of 1000 mg IV every 12 h to support efficacy parameters. Regrettably, the explanation by the committee opinion on this subject simply reiterates the false assumptions about the vancomycin concentration–effect relationship provided by the authors of the original PK paper, discussion points we disapprove of as they do not conform to reasonable dose selection based on modern standards of antimicrobial PK/PD [3]. The authors described the “therapeutic level” to be a level between 10 and 40 mg/L, a largely generalized and a potentially misleading depiction of concentrations associated with vancomycin’s therapeutic effects that, when poorly translated to clinical care, may result in sub-therapeutic dosing or increasing the risk of adverse drug events [6]. One of the most common non-severe reactions of vancomycin includes flushing, which can be overcome with antihistamines and a slower infusion time but is more likely to occur when higher doses are administered [7]. More serious adverse events include nephrotoxicity that impacts both the adult and neonatal populations. Le and colleagues conducted a large population-based PK analysis using 1576 concentrations collected from 680 pediatric subjects and found that an AUC of ≥800 mg·h/L and trough concentrations of ≥15 mg/L were independently associated with an increased risk of acute kidney injury (AKI) [8]. In two additional papers, Bhargava, et al. and Dawoud et al. analyzed the incidence of AKI and also found that there is an association between the rate of AKI and elevated vancomycin concentration, and while the overall incidence was low, it was found to be highest in the neonate population with trough levels greater than 15 mg/L [9,10]. Limiting the intrapartum dose to a maximum of 1250 mg is justified simultaneously for the maternal and neonatal population at our proposed targets when an empiric approach to dosing is considered. In the context of dosing a neonate with vancomycin on the first day of life, the 1250 mg dose also ensures that the risk of overdose beyond the recommended 30 mg/kg/day is limited to below 10% for the majority of the GA categories and birth times investigated.

The second stipulation to support the high dose approach by #797 aims to minimize the risk of resistance development. This perception is also strongly echoed by the authors of the PK study of intrapartum vancomycin where they quote the “at the time current” antimicrobial guidelines stating that “minimum serum vancomycin trough concentrations should always be maintained above 10 mg/L to avoid development of resistance”, a remark we contend and one which we must further elucidate [11]. Examination of the guideline references evaluating data derived from clinical observations shows that, indeed, vancomycin concentrations lower than 10 mg/L may be related to the emergence of resistant staphylococcus after prolonged (several days to weeks) vancomycin exposure, but there are no implications found in those papers that they also influence resistance occurrence in streptococcal strains [12,13,14]. Additionally, previous studies have helped to identify risk factors that may contribute to vancomycin resistance emergence, such as previous MRSA colonization, hemodialysis dependence, long-term use of vancomycin, hospitalization in ICU, and use of indwelling devices, but there are no reports that describe intrapartum vancomycin prophylaxis to similarly influence the same [15]. On the contrary, there is evidence to confirm that the use of intrapartum prophylaxis does not meaningfully impact observed antibiotic resistance which, when combined with the lack of any conclusive evidence from clinical observations, invalidates the presumption that concentrations below 10 mg/L during standard intrapartum prophylaxis with vancomycin drive resistance development in streptococci [16,17,18].

The primary limitation of this work, we believe, is the use of an efficacy related PD index to assess for PTAs that were derived from animal experiments lacking validation in the actual intrapartum clinical context. Additionally, the simulation in our study is based on the analysis of published PK information, where a model of substantial complexity is used to analyze data of sparse sampling design. Last, it is also important to note though that our ability to identify significant covariate relationships in this work may also be hindered by the small sample size, the sparse sampling design, and the overall complexity of the structural model.

## 4. Methods

### 4.1. Study Design

The protocol of the PK study, which included 30 patients, and the subsequent interpretation of the measured vancomycin concentrations guiding the #797 recommendations for intrapartum vancomycin administration have been previously described [3]. Briefly, pre- and full-term mothers entering labor who presented with high risk of penicillin allergy and positive GBS culture resistant to clindamycin/erythromycin or with an unknown GBS susceptibility profile were included. A weight-based vancomycin dose of 20 mg/kg (maximum of 2000 mg single dose) every 8 h was given to all consenting mothers. Maternal and cord blood samples were collected at the time of delivery. Clinical and demographic data were also collected for each mother and newborn. Since our analysis involved only the use of previously published data, no local approval by IRB was required.

### 4.2. Pharmacokinetic Analysis

Due to the sparse sampling of the original design a two-compartment model with zero-order infusion and first-order intercompartmental transfer and elimination was fitted to the data with the Pmetrics software program [19]. The structure of the model equations used to characterize the PK profile of intrapartum administration of vancomycin is shown in Figure 2. *X_1_* is the amount of drug in the maternal compartment, *X*_2_ is the amount of drug in the neonatal compartment, *k_el_* is the maternal elimination rate constant (h^−1^), *v_mat_* and *v_fet_* are the volume of the maternal and fetal compartment (L), and *k*_mf_ and *k*_fm_ are inter-compartmental transfer rate constants between the maternal and fetal compartments (h^−1^). *RESID* is the amount of vancomycin remaining in the neonatal compartment at the time of delivery (mg), *R*_1_ is the zero-order drug input rate into the maternal compartment (mg/hour), and *R*_2_ is the control for transfer between maternal and fetal compartments constrained to 1 (before delivery) or 0 (after delivery). Tested covariates included maternal body weight. Criteria for inclusion of the covariate in the final model was based on an improvement in the log-likelihood value (*p* < 0.05) and/or improvement in the goodness-of-fit plots. Coefficients for the assay error polynomial model with a gamma function (γ) was determined from the data using the *ERRrun()* function in Pmetrics. The fit of the model to the data was assessed by mean weighted prediction error (a measure of bias), by bias-adjusted mean weighted squared prediction error (a measure of precision), coefficient of determination (r^2^) of the linear regression of the observed and predicted values, and by visual inspection. In addition, log likelihood ratio tests and the Akaike information criterion (AIC) were also considered during model selection.

### 4.3. Monte Carlo Simulation

Monte Carlo simulations for 1000 subjects using the final model for the population, then for 1000, 1250, 1500, 1750, and 2000 mg intrapartum vancomycin regimens every 8 h and for the maximum duration of a 5-dose course and for possible birth times up to 4-hour intervals was performed (available in Appendix A). The results labeled by the term population represent the combined output calculated from the predictions based on the individual maternal–fetal dosing records (1000 virtual patients simulated for each) as published in the original PK study. The framework provided by the ID-ODS^TM^ (Individually Designed Optimum Dosing Strategies, http://www.optimum-dosingstrategies.org/, accessed on 10 October 2022) application was used to carry out these analyses. ID-ODS^TM^ is a simulation tool that supports probabilistic forecasting and Bayesian adaptive feedback powered by the R^®^ software (version 3.6.2; Institute for Statistics and Mathematics, http://www.r-project.org/, accessed on 10 October 2022) with an extensive model library built from published population PK models [20]. Due to the limitations of the sampling strategy in the original PK study, estimation of the neonatal clearance was not possible. Subsequently, distribution of the parameter values from a validated population PK model established in neonates was used during simulations for GA of 33 to 40 weeks [5]. Neonatal body weight and creatinine values for the simulation were generated using previously published data [21]. Vancomycin efficacy and toxicity indices unique to maternal and neonatal populations were evaluated for PTA. The current guidelines on vancomycin therapeutic monitoring recommend using the AUC with or without the MIC (frequently assumed to be 1 mg/L) as ratio as the most appropriate PK/PD target of interest that accurately describes the therapeutic effectiveness and safety of vancomycin treatment [22]. Although clinical outcome data is lacking, efforts to quantify this relationship for streptococci in standard mouse models of infection demonstrated that the AUC/MIC ratio was an important PD parameter predictive of efficacy [23]. Lee et al. found that the AUC/MIC ratio and the magnitude for total drug required for maximal effect at 24 h for S. aureus was 399, a value similar to the lower end of the therapeutic range recommended by the current vancomycin dosing guidelines, while for Streptococcus to achieve a similar outcome, a 10-fold lower index of 39.1 was identified [23]. These magnitudes of the PD index associated with the maximum effect in mice most often correspond to human exposure necessary to treat life-threatening infections where the bacterial density is often expected to be high (>9 log_10_ CFU/g of tissue) and is unlikely to be required to achieve clinical success in the current context when only intrapartum GBS prophylaxis is considered [24]. Under maternal conditions, GBS colonization is generally known to be light, with most of the cases limited to bacterial density of 4 log_10_ to 6 log_10_ CFUs, and resemble that of the average density expected in acute bacterial skin and skin structure infections (ABSSSI) [25,26,27,28]. Taking into consideration the mechanism of exposure of the neonate to GBS by the ascending route in utero or by contamination during passage via the birth canal (and the low bacterial densities to be encountered) supports a rationale for choosing a PD target for intrapartum prophylaxis that is more closely aligned with results of animal models supporting ABSSSI-related dose selection versus the more aggressive approach intended for life-threatening infections. Additionally, based on clinical observations, the very rapid reduction of vaginal GBS colony counts following the administration of intrapartum vancomycin also supports the hypothesis against the need to target an aggressive PD index [29]. For ABSSSI, the exposure target associated with a static effect to a 1 log CFU reduction in colony counts from baseline in the thigh infection model has been shown to be predictive of efficacy in subsequent ABSSSI clinical trials [30]. Accordingly, we favored the robust AUC/MIC ratio of 39.1 in our analysis for efficacy and at the susceptibility breakpoint of GBS to vancomycin of 1 mg/L [31]. The pre-defined PD target for safety was the assessment of the probability of achieving vancomycin exposure of the AUC > 600 (maternal/neonatal toxicity threshold), and the more rigorous AUC > 800 mg·h/L and concentrations above 15 mg/L before the 4th dose (neonatal specific toxicity threshold) [22]. To further characterize vancomycin exposure the newborn population is subjected to, the amount of drug in the neonate at the time of birth was also calculated and compared with maximum dosing recommendations (30 mg/kg/day) of vancomycin in the treatment of complicated neonatal infections when administered on the first day of life [32].

## Figures and Tables

**Figure 3 antibiotics-12-00032-f003:**
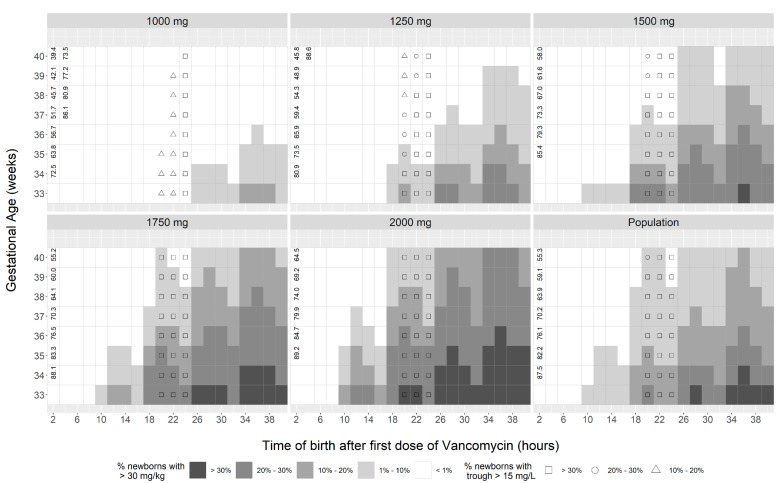
Newborn probability of vancomycin accumulation in excess of the 30 mg/kg dose cutoff. The shapes inside the tiles at 20, 22, and 24 h birth times represent probability of vancomycin concentrations > 15 mg/L after 3 doses administered. Numerical values in the boxes for birth times of 4 h or less indicate the probability of successful target achievement at the efficacy index of AUC > 39.1 mg·h/L not reaching the threshold of 90% (only <90% values are shown, for all others the PTA is >90%). As an example, a newborn with GA of 35 weeks born at 22 h after initiation of intrapartum prophylaxis to a mother who received 3 doses of the 2000 mg prophylactic regimen is at 20% to 30% risk of accumulating vancomycin in excess of the 30 mg/kg cutoff and is at >90% chance of achieving the target efficacy index of AUC > 39.1 mg·h/L and has the probability > 30% to have a serum trough concentration of >15 mg/L 24 h after initiation of intrapartum prophylaxis.

**Figure 4 antibiotics-12-00032-f004:**
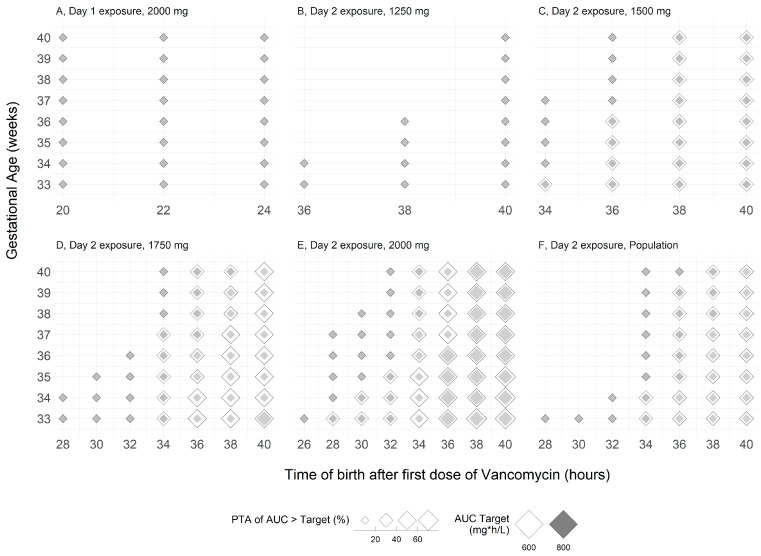
(**A**–**F**) Population and dose-wise newborn target attainment greater than the threshold of 10% for the toxicity indices (only PTAs > 10% are shown, for all other birth times and GAs the PTA is <10%). As an example, a newborn with GA of 35 weeks born at 34 h after initiation of intrapartum prophylaxis to a mother who received 5 doses of the 2000 mg prophylactic regimen is at 40% to 60% risk of exposure in excess of the AUC > 600 mg·h/L and has a 10% to 20% chance of exposure in excess of the AUC > 800 mg·h/L cutoff.

**Figure 5 antibiotics-12-00032-f005:**
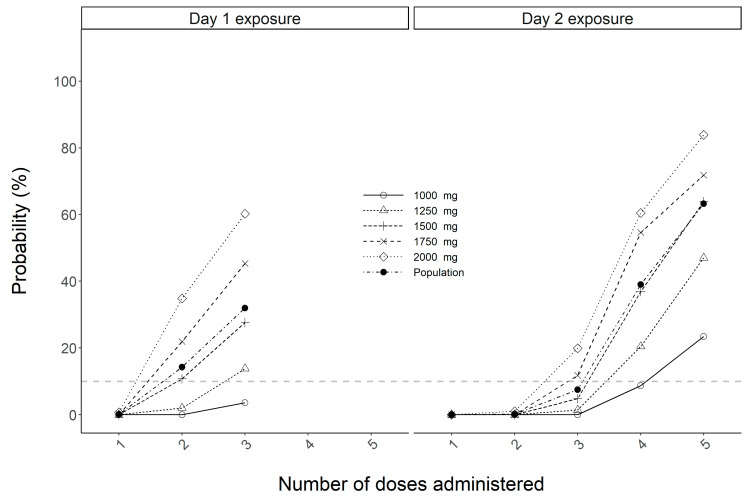
Maternal probability of target attainment for the toxicity index of the AUC above 600 mg·h/L. Broken grey horizontal line represents the cutoff of 10%.

## Data Availability

The data presented in this study are available in Appendix A.

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
