# Peer review of "Sorting Out the Risks and Benefits of the #797 Recommended Intrapartum Vancomycin Dosing Approach"

_antibiotics, 2022, doi:10.3390/antibiotics12010032_

Round 1

Reviewer 1 Report

Introduction

Line 33: Full name of ACOG should be written when it is used for first time.

Lines 38-39: Please, revise the English language of the sentence “If case of low-risk allergy to penicillin a first generation cephalosporins is recommended versus for serious reactions clindamycin and vancomycin are suggested.”

Lines 42-45: The authors wrote about “pitfall associated with both of the studies however, is the author’s unconventional interpretation ….”. The statement is too strong and required a revision. Written in such way, without any evidences for this statement, the readers are not convinced that the statement is true.

Material and methods: The section is well described and clearly explained.

If it is possible, it will be good if the authors prepare a supplementary file with the data which were used for Monte Carlo simulations.

Results and discussion are well presented in the text. The figures are unusual and their clarity can be improved by better explanations under the figures.

The discussion presents logical interpretation of data. The message is clear and properly reflects the data. The manuscript can be published because it has additive value to information about the possibilities for treatment of uterine infections with vancomycin.

Reviewer 2 Report

This study is fairly well designed and written.

Please see the attached file with comments to be addressed by the authors
